# Raising awareness of potential biases in medical machine learning: Experience from a Datathon

Harry Hochheiser[1]*, Jesse Klug[2], Thomas Mathie[3], Tom J. Pollard[4], Jesse D. Raffa[4], Stephanie L. Ballard[5], Evamarie A. Conrad[5], Smitha Edakalavan[1], Allan Joseph[6,7], Nader Alnomasy[5,8], Sarah Nutman[3], Veronika Hill[5], Sumit Kapoor[3], Eddie Pérez Claudio[1], Olga V. Kravchenko[9], Ruoting Li[3], Mehdi Nourelahi[1], Jenny Diaz[5], W. Michael Taylor[3], Sydney R. Rooney[10], Maeve Woeltje[3], Leo Anthony Celi[4,11,12], Christopher M. Horvat[3]

**1** Department of Biomedical Informatics, University of Pittsburgh, Pittsburgh, Pennsylvania, United States of America, **2** UPMC Intensive Care Unit Service Center, UPMC, Pittsburgh, Pennsylvania, United States of America, **3** Department of Critical Care Medicine, University of Pittsburgh, Pittsburgh, Pennsylvania, United States of America, **4** MIT Laboratory for Computational Physiology, Institute for Medical Engineering and Science, Massachusetts Institute of Technology, Cambridge, Massachusetts, United States of America, **5** Health Informatics, School of Health and Rehabilitation Sciences, University of Pittsburgh, Pittsburgh, Pennsylvania, United States of America, **6** Division of Critical Care Medicine, Cincinnati Children's Hospital Medical Center, Cincinnati, Ohio, United States of America, **7** Department of Pediatrics, University of Cincinnati College of Medicine, Cincinnati, Ohio, United States of America, **8** College of Nursing, Medical Surgical Department, University of Ha'il, Ha'il, Saudi Arabia, **9** Department of Family and Community Medicine, University of Pittsburgh, Pittsburgh, Pennsylvania, United States of America, **10** Division of Cardiology, Department of Pediatrics, Children's Hospital of Pittsburgh, University of Pittsburgh, Pittsburgh, Pennsylvania, United States of America, **11** Division of Pulmonary, Critical Care and Sleep Medicine, Beth Israel Deaconess Medical Center, Boston, Massachusetts, United States of America, **12** Department of Biostatistics, Harvard T.H. Chan School of Public Health, Boston, Massachusetts, United States of America

* harryh@pitt.edu

**Data availability statement:** Datathon materials, including introductory presentations,

## Abstract

**Objective:** To challenge clinicians and informaticians to learn about potential sources of bias in medical machine learning models through investigation of data and predictions from an open-source severity of illness score.

**Methods:** Over a two-day period (total elapsed time approximately 28 hours), we conducted a datathon that challenged interdisciplinary teams to investigate potential sources of bias in the Global Open Source Severity of Illness Score. Teams were invited to develop hypotheses, to use tools of their choosing to identify potential sources of bias, and to provide a final report.

**Results:** Five teams participated, three of which included both informaticians and clinicians. Most (4/5) used Python for analyses, the remaining team used R. Common analysis themes included relationship of the GOSSIS-1 prediction score with demographics and care related variables; relationships between demographics and outcomes; calibration and factors related to the context of care; and the impact of missingness. Representativeness of the population, differences in calibration and model performance among

instructions, and submissions from each of the five teams can be found at https://doi.org/10.5281/zenodo.13962037.

**Funding:** TP and LAC are supported by Nationals Institute of Health Grant OT2OD032701. LAC is also funded by NIH Grants R01 EB017205 and DS-I Africa U54 TW012043-01 and the National Science Foundation through ITEST #2148451. The funders had no role in study design, data collection and analysis, decision to publish, or preparation of the manuscript.

**Competing interests:** The authors have declared that no competing interests exist.

groups, and differences in performance across hospital settings were identified as possible sources of bias.

**Discussion:** Datathons are a promising approach for challenging developers and users to explore questions relating to unrecognized biases in medical machine learning algorithms.

## Author summary

Members of economically or socially disadvantaged groups are at risk of being adversely impacted by biased medical machine learning models. To avoid these undesirable outcomes, developers and users must understand the challenges involved in identifying potential biases. We conducted a datathon aimed at challenging a diverse group of participants to explore an open-source patient severity model for potential biases. Five groups of clinicians and informaticians used tools of their choosing to evaluate possible sources of biases, applying a range of analytic techniques and exploring multiple features. By engaging diverse participants with hands-on data experience with meaningful data, datathons have the potential to raise awareness of potential biases and promote best practices in developing fair and equitable medical machine learning models.

## Introduction

Effective clinical artificial intelligence (AI) and machine learning (ML) tools must be based on robust models tested to minimize undesirable or unexpected outcomes. Construction of such models requires examination of underlying assumptions, data distributions, and model output for potentially subtle biases that might penalize sub-populations, leading to disproportionate allocation of resources or adverse outcomes. Recent efforts have demonstrated potential adverse impacts of biased models for resource allocation, [1] computer vision applications for diagnosis, [2,3] and models for risks of urinary tract infections, [4] among other applications. Although reporting tools, [5,6] checklists, [7] and bias exploration libraries [8–10] provide some assistance, the management of bias in medical ML projects is still a largely manual task, requiring exploration of data and testing of specified hypotheses. Effective efforts will likely rely on multiple perspectives, particularly bridging gaps between clinicians with domain knowledge necessary for generation of hypothetical sources of bias and analysts with statistical and programming skills necessary to test those hypotheses. To raise awareness of these issues among a cohort of individuals involved in developing medical AI tools, we conducted and participated in a datathon that challenged groups of informaticians and clinicians to generate hypotheses regarding potential biases in the Global Open Source Severity of Illness Score (GOSSIS-1) prediction algorithm [11].

Many studies have identified medical machine learning models that appear to offer predictions or classifications based on an individuals' identity, rather than their medical condition [3,12,13]. Although these biases are often manifested as differences in model performance across racial groups, they arise in many different ways. Many algorithms have explicitly used race as a predictive variable. Some, such as the UTICalc model for predicting probabilities of urinary tract infections in children, [4] have been updated in revised versions built without race as a feature [14,15]. Computer vision-based diagnostic tools have also been a source of bias, as models trained only on datasets involving lighter skin have been observed

to have drastically reduced performance for darker skin [2,3]. Complex interplays between demographic variables and AI models can lead to bias as well. Obermeyer, et al.'s analysis of a model for prioritization of critical care resources noted that the inclusion of medical-care utilization patterns as a predictive feature led to a bias against African-Americans, who might tend to have lower income and therefore lower levels of utilization [1]. These and other examples, including general investigations of the impact of race in machine learning models, illustrate both the possible undesired outcomes of bias models and the challenges inherent in mitigating biases [16,17].

A variety of approaches have been developed to encourage the use of best practices for reducing risks of bias. Reporting tools such as the Prediction model Risk of Bias Assessment (PROBAST) [5,18] and the Transparent Reporting of a multivariable prediction model for Individual Prognosis or Diagnosis (TRIPOD), [19,20] provide guidance respectively on how modeling efforts might be examined for bias and how appropriate elements of a modeling effort might be reported. These tools have recently been expanded to explicitly cover AI models, through the PROBAST-AI and TRIPOD-AI extensions [6,21]. Several other similar reporting guidelines [22–24] and strategies for reducing bias [7,25] have been proposed.

These efforts are complicated by the absence of clear definitions of bias or related terms such as model fairness. Beyond the relatively basic definitions of group fairness (disadvantaged groups should be treated similarly to advantaged groups) and individual fairness (similar individuals should be treated similarly), [26] a range of metrics provide different perspectives on how fairness and bias might be assessed [26]. Software toolkits, including Microsoft's Fairlearn, [8] AI Fairness 360, [10] and Aequitas [9] provide automated or semi-automated audits for compliance with multiple fairness metrics, together with guidance as to how biases might be ameliorated. However, these measures should be used carefully, as they may have negative implications for model performance [15,27].

Although frameworks for investigating sources of bias, reporting guidelines, and audit toolkits are useful tools for reducing unintended biases, their impacts will be limited by adoption. Educational efforts will be needed to inform researchers, data scientists, and clinicians regarding potential source of biases in machine learning models and strategies for identifying and mitigating those biases. As a demonstration of a potentially promising educational approach, we conducted and participated in a bias in clinical AI/ML datathon in February 2024. Building off of prior demonstrations of the educational utility of datathons, [28–31] our datathon challenged participants to work in teams to identify possible biases in a recently-published machine learning model.

Participants in the datathon examined the Global Open Source Severity of Illness Score (GOSSIS-1) prediction algorithm [11]. The GOSSIS-1 score was developed in response to the generalizability and performance degradation problems impacting earlier severity of illness scores [32]. The development of the GOSSIS-1 score was based on datasets from the eICU database of over 200,000 ICU admissions in the US [33] and a comparable database of more than 60,000 ICU admissions in Australia and New Zealand [34] to develop a new model and demonstrate a proposed methodology for developing globally-applicable severity scores. Although logistic regression models based on demographics, labs, vitals, and APACHE score predictions led to models with good performance and calibration, [11] potential biases in the data used to train the GOSSIS model have not yet been thoroughly studied. Our datathon challenged groups of informaticians and clinicians to generate hypotheses regarding potential sources of bias in the GOSSIS-1 training data, thus providing participants with the opportunity to gain hands-on experience in dealing with understanding bias in clinical AI models while potentially providing insights into the GOSSIS-1 data.

## Materials and methods

The datathon was intended primarily to challenge emerging researchers to consider questions relating to bias in predictive models. Participants (all of whom are co-authors) included medical residents and fellows, students, faculty, and staff from the University of Pittsburgh and UPMC. The datathon was held virtually, over 1.5 working days in February 2024. The first day included background presentations on the GOSSIS model and data, an introduction to bias in machine learning, a description of the Prediction model Risk Of Bias ASsessment Tool (PROBAST) bias reporting framework, [5] and a description of the datathon's goals and judging criteria. From 1pm onwards, participants met and conducted analyses. The second day of the datathon started with one hour of short talks about ML/AI in medicine, followed by three hours of open time for working on assignments.

As the goal of the datathon was to invite participation from a brorad range of participants with differing levels of experience in analyzing clinical data sets, we provided multiple ways to participate. Teams had three options for accessing data to be used in the Datathon. Two datasets from PhysioNet were available: 1) the Women in Data Science Datathon 2020 dataset (WIDS2020) [35,36] was available both in raw form and in a version that could be analyzed using ChatGPT Advanced Data Analysis Features, and 2) those who had completed appropriate Physionet credentialing were eligible to use the eICU subset of the GOSSIS-1 dataset [36, 37] containing the US data used in developing the GOSSIS-1 score. A third dataset was a synthetic derivative of data used to implement the GOSSIS-1 score at UPMC. Participants who chose either the eICU dataset or the synthetic UPMC data set were not able to use the ChatGPT function directly with the data, due to restrictions on data sharing, but could still use generative AI tools to assist with code writing.

In addition to selecting one of several data options, participating teams were allowed to use tools of their choice for analyzing data and presenting results. They were asked to develop hypotheses regarding potential sources of bias, using PROBAST questions regarding predictors, outcomes, and analysis as a framework for guiding hypothesis development. A grading rubric was provided and teams were instructed to provide a report detailing their hypotheses, features used, indications of any supplemental data, descriptions of analyses conducted, presentation of results, and descriptions of conclusions. Submissions were analyzed to extract tools used and to categorize analysis approaches used and resulting conclusions into common themes. Datathon materials, including introductory presentations, instructions, and submissions from each of the five teams can be found at https://doi.org/10.5281/zenodo.13962037.

*Ethics Statement:* The University of Pittsburgh Institutional Review Board approved this work (STUDY24010010: Critical Care Datathon) as exempt research. All participants in the datathon are included as co-authors.

## Results

The 26 individuals who registered for the datathon were initially divided into five teams of roughly comparable size (four teams of five participants each and one team with six participants). As several registered participants either did not participate at all or did not continue to completion, final team sizes ranged from 1 to 5 members. Three of the teams included both clinicians and informaticians. One of the remaining teams had two members, both clinicians; the other team included one participant, an informatician. Slightly over one-half (9/16) of the participants had a clinical background. Detailed information on participants' experience with coding or data analytics was not collected.

Of the five teams, two used the synthetic UPMC dataset, two used the WIDS2020 dataset, and one team used both. No teams used the eICU dataset. Python was the preferred analysis tool, selected by four of the five teams. The remaining team used R. At least one team used a GUI-based tool for exploratory data analysis, and one team acknowledged using ChatGPT to assist with coding. Most (4) of the teams used Jupyter notebooks, one team used R.Four of the teams formatted their final report as a PDF document; the remaining team created a slide presentation. Teams used a variety of approaches for data presentation, including bar graphs, AUROC/AUPRC graphs, calibration plots, and heatmaps.

Participating teams used a variety of approaches to examine potential sources of bias. Three of the five teams provided explicitly stated hypotheses; four of the teams referred to PROBAST questions either in the formulation of hypotheses or in the presentation of their conclusions. Age, race/ethnicity, and mortality were the most frequently examined features, considered by four of the five teams. A complete list of features or data characteristics and the frequency of their use by the teams is given in Table 1.

Analysis approaches fell into three broad categories (Table 2). All 5 groups explored potential biases in GOSSIS scores including analyses of the relationships between predictions and demographics, context of care (hospital/ICU types), admission source, and exploration of calibration of GOSSIS predictions across ethnic groups. All groups also explored the impact of demographics (race, ethnicity, sex/gender, and age), including associations with outcomes, hospital type, length of pre-ICU hospital stay, feature importance, and calibration, along with comparison of the patient population relative to the general population (as indicated by census data).

Teams used a variety of graphical and tabular approaches to present results. One team used an external data source (US Census Bureau information). Only one of the teams used any inferential statistical tests in their analyses. Although none of the teams found clear instances of bias, several areas of concern were identified, including possible biases in missingness of the data (n=2), representativeness of the population (2) , differences in calibration (3) and model performance among groups (3), and differences in performance across hospital settings (2).

**Table 1. Data features examined for potential biases.**

| Feature/characteristic | # of teams (n=5) |
|---|---|
| GOSSIS predicted score | 5 |
| Age | 4 |
| Race/ethnicity | 4 |
| Mortality | 4 |
| Type of ICU/Hospital | 3 |
| Missingness | 3 |
| APennsylvaniaCHE score | 3 |
| Calibration | 3 |
| Sex/gender | 2 |
| Comorbidities | 1 |
| Length of pre-ICU hospitalization | 1 |
| Data entry errors | 1 |
| Ventilation status | 1 |
| Feature importance | 1 |
| Readmission | 1 |
| Impact of imputation | 1 |
| ICU Admissions source | 1 |

**Table 2. Analysis topics.**

| Analysis Category | Subcategory | # of teams (n=5) |
|---|---|---|
| GOSSIS Score | Relationship with admission source (1), demographics (2), hospital/ICU type (2), admission source(1); association of calibration with ethnicity (3) | 5 |
| Demographics | Relationship with GOSSIS score (1), death/survival (1); Association with comorbidities (1), hospital type (1), length of pre-icu hospital stay (1), ventilation use (1), feature importance (1), and calibration (3) ; general distribution patterns (1) | 5 |
| Missingness | Association with demographics (1), GOSSIS score (1), hospital/ICU type (1); Impact of imputation on model performance (1) | 3 |

## Discussion

Despite significant awareness of the potential challenges and harms associated with inappropriately-biased medical AI/ML models, appropriate strategies for identifying and addressing bias are not as well understood. A range of techniques have been proposed to identify and ameliorate bias [38,39] leading to the development of software libraries for measuring potential biases [8,10] and decision tools designed to help researchers choose the most suitable fairness metrics for a given situation [9]. Several broad frameworks for addressing bias have been proposed, with varying levels of specificity [7,25,40,41]. Given the numerous possible sources of interactions, the complexity of biases they may introduce, and the need for additional data to provide context, identification of potential biases in machine learning models will likely remain an ongoing challenge requiring thoughtful examination of the models and the data. Educating ML developers and clinicians about the potential dangers of biased models is perhaps the most promising means of addressing this challenging problem.

Our datathon provides a model for engaging informaticians and clinicians in collaborative efforts to explore potential sources of undesired biases in machine learning models. Mixed groups involving participants with a range of clinical and technical experience were able to come together to quickly begin exploration of multiple hypotheses. Although the five teams identified different questions and used varying approaches to address those questions, interactions were collaborative and productive. The open-ended nature of the tasks and the straightforward data involved likely helped informaticians to generate relevant questions, despite their lack of clinical expertise. Similarly, flexibility of tools enabled clinicians who might not be well-versed in programming or statistical applications to use more familiar tools (although some clinicians were proficient R/Python programmers). Anecdotal observation of the groups in action suggested that exchanges between clinicians and informaticians enhanced participants' understanding of the complexities of the questions.

Teams shared a common focus on topics considered in their analyses and the features used to conduct those analyses. The relative lack of inclusion of statistical tests and external data sets (1 team for each) suggests that discussion of rigorous methods for quantifying the likelihood of bias and the potential utility of including additional data sources for contextualizing potential sources of bias might be useful topics for related training. Most (four of five) teams used the PROBAST model to frame their analyses. Similar detailed questions designed to guide analyses or decision tools such as the fairness tree [9] might be helpful for guiding participants in future datathons, and for generally helping developers address questions of potential bias. None of the teams identified strong evidence for troubling bias. Given the limited available time, this is not surprising. However, they identified potentially concerning areas

worthy of further consideration, representativeness of data, calibration, model performance, and the potential impact of context of care.

Beyond the framework provided by the PROBAST questions, [5] participants were given very little structure or guidance in developing hypotheses and exploring data. The open-ended nature of the task reflects the lack of structured approaches for conducting these analyses, and may have encouraged some creativity in addressing this problem. Future datathons might use a modified approach providing additional scaffolding in the form of structured questions, potentially encouraging the use of more formalized processes for identifying and testing hypotheses regarding potential biases.

Additional future enhancements might involve evaluation of the impact of the datathon and the insertion of datathon activities in a broader educational context. Content-based evaluations might address participants' understanding of bias, fairness, and other related concepts. Self-efficacy questions might use a quasi-experimental approach to explore any changes in participants' perceptions of their ability to address these topics. Datathon activities might also be embedded in more formal classes, providing opportunities for theoretical introduction of fairness and bias, including definitions of specific measures and discussions of how they might be used. Although this additional background would likely be of use to many participants, the mixing of perspectives associated with a datathon involving both clinicians and data scientists might be difficult to achieve in more formal educational settings.

## Conclusion

Identification and reduction of inappropriate biases of medical machine learning models are increasingly important goals, likely requiring collaboration between developers and clinical users of those models. Our February 2024 datathon provides a model for using team-based investigation of meaningful datasets to explore hypotheses and identify potential sources of bias in need of further consideration. We hope to use subsequent datathons to explore possible systematic approaches for identifying biases in both underlying data and models trained using those data and improve methods to mitigate these biases to achieve equitable outcomes for all patients.

## Author contributions

**Conceptualization:** Harry Hochheiser, Thomas Mathie, Tom Pollard, Jesse Raffa, Leo Anthony G. Celi, Christopher M. Horvat.

**Data curation:** Harry Hochheiser, Jesse Klug, Thomas Mathie, Tom Pollard, Jesse Raffa, Leo Anthony G. Celi, Christopher M. Horvat.

**Formal analysis:** Harry Hochheiser, Stephanie L. Ballard, Evamarie A. Conrad, Smitha Edakalavan, Allan M Joseph, Nader Alnomasy, Sarah Nutman, Veronika Hill, Sumit Kapoor, Eddie Perez Claudio, Olga V Kravchenko, Ruoting Li, Mehdi Nourelahi, Jenny Diaz, W. Michael Taylor, Sydney Rooney, Maeve Woeltje.

**Funding acquisition:** Harry Hochheiser, Leo Anthony G. Celi, Christopher M. Horvat.

**Investigation:** Harry Hochheiser, Tom Pollard, Stephanie L. Ballard, Evamarie A. Conrad, Smitha Edakalavan, Allan M Joseph, Nader Alnomasy, Veronika Hill, Sumit Kapoor, Eddie Perez Claudio, Olga V Kravchenko, Ruoting Li, Mehdi Nourelahi, Jenny Diaz, W. Michael Taylor, Sydney Rooney, Maeve Woeltje, Leo Anthony G. Celi, Christopher M. Horvat.

**Methodology:** Harry Hochheiser, Tom Pollard, Jesse Raffa, Stephanie L. Ballard, Evamarie A. Conrad, Smitha Edakalavan, Allan M Joseph, Nader Alnomasy, Sarah Nutman, Veronika

Hill, Sumit Kapoor, Eddie Perez Claudio, Olga V Kravchenko, Ruoting Li, Mehdi Noure-lahi, Jenny Diaz, W. Michael Taylor, Sydney Rooney, Maeve Woeltje, Leo Anthony G. Celi, Christopher M. Horvat.

**Project administration:** Harry Hochheiser, Tom Pollard, Jesse Raffa, Leo Anthony G. Celi, Christopher M. Horvat.

**Resources:** Harry Hochheiser, Jesse Klug, Thomas Mathie, Tom Pollard, Jesse Raffa, Stephanie L. Ballard, Evamarie A. Conrad, Smitha Edakalavan, Allan M Joseph, Nader Alnomasy, Sarah Nutman, Veronika Hill, Sumit Kapoor, Eddie Perez Claudio, Olga V Kravchenko, Ruoting Li, Mehdi Nourelahi, Jenny Diaz, W. Michael Taylor, Sydney Rooney, Maeve Woeltje, Leo Anthony G. Celi, Christopher M. Horvat.

**Software:** Stephanie L. Ballard, Evamarie A. Conrad, Smitha Edakalavan, Allan M Joseph, Nader Alnomasy, Sarah Nutman, Veronika Hill, Sumit Kapoor, Eddie Perez Claudio, Olga V Kravchenko, Ruoting Li, Mehdi Nourelahi, Jenny Diaz, W. Michael Taylor, Sydney Rooney, Maeve Woeltje.

**Supervision:** Harry Hochheiser, Jesse Raffa, Leo Anthony G. Celi, Christopher M. Horvat.

**Visualization:** Stephanie L. Ballard, Evamarie A. Conrad, Smitha Edakalavan, Allan M Joseph, Nader Alnomasy, Sarah Nutman, Veronika Hill, Sumit Kapoor, Eddie Perez Claudio, Olga V Kravchenko, Ruoting Li, Mehdi Nourelahi, Jenny Diaz, W. Michael Taylor, Sydney Rooney, Maeve Woeltje.

**Writing – original draft:** Harry Hochheiser, Tom Pollard.

**Writing – review & editing:** Harry Hochheiser, Jesse Klug, Thomas Mathie, Tom Pollard, Jesse Raffa, Stephanie L. Ballard, Evamarie A. Conrad, Smitha Edakalavan, Allan M Joseph, Nader Alnomasy, Sarah Nutman, Veronika Hill, Sumit Kapoor, Eddie Perez Claudio, Olga V Kravchenko, Ruoting Li, Mehdi Nourelahi, Jenny Diaz, W. Michael Taylor, Sydney Rooney, Maeve Woeltje, Leo Anthony G. Celi, Christopher M. Horvat.

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
