## [Decision Letter · Decision Letter 0]

16 Jan 2025

PDIG-D-24-00484Raising awareness of potential biases in medical machine learning: Experience from a DatathonPLOS Digital Health Dear Dr. Hochheiser, Thank you for submitting your manuscript to PLOS Digital Health. After careful consideration, we feel that it has merit but does not fully meet PLOS Digital Health's publication criteria as it currently stands. Therefore, we invite you to submit a revised version of the manuscript that addresses the points raised during the review process. Please submit your revised manuscript within 60 days Mar 17 2025 11:59PM. If you will need more time than this to complete your revisions, please reply to this message or contact the journal office at digitalhealth@plos.org. Please include the following items when submitting your revised manuscript:* A rebuttal letter that responds to each point raised by the editor and reviewer(s). You should upload this letter as a separate file labeled 'Response to Reviewers'. This file does not need to include responses to any formatting updates and technical items listed in the 'Journal Requirements' section below.* A marked-up copy of your manuscript that highlights changes made to the original version. You should upload this as a separate file labeled 'Revised Manuscript with Track Changes'.* An unmarked version of your revised paper without tracked changes. You should upload this as a separate file labeled 'Manuscript'. If you would like to make changes to your financial disclosure, competing interests statement, or data availability statement, please make these updates within the submission form at the time of resubmission. Guidelines for resubmitting your figure files are available below the reviewer comments at the end of this letter. We look forward to receiving your revised manuscript. Kind regards, Wisit Cheungpasitporn, MDAcademic EditorPLOS Digital Health Wisit CheungpasitpornAcademic EditorPLOS Digital Health Leo Anthony CeliEditor-in-ChiefPLOS Digital Healthorcid.org/0000-0001-6712-6626 **Journal Requirements:**

1. We ask that a manuscript source file is provided at Revision. Please upload your manuscript file as a .doc, .docx, .rtf or .tex.

 **Additional Editor Comments (if provided):** I find this manuscript addressing bias awareness in medical machine learning through a datathon experience requires substantial revision despite its promising and relevant topic. The primary concerns from reviewers include structural issues, with the paper lacking a proper literature review either as a standalone section or within the introduction, methodology content appearing in incorrect sections, and mathematical inconsistencies regarding team sizes (16 participants cannot form 5 equal teams). The introduction and discussion sections need significant expansion to provide stronger context about bias in medical ML and better support for the findings, while the methodology requires clearer description of analytical approaches and proper placement of content.**Reviewers' Comments:** Reviewer's Responses to Questions

**Comments to the Author**

1. Does this manuscript meet PLOS Digital Health’s publication criteria? Is the manuscript technically sound, and do the data support the conclusions? The manuscript must describe methodologically and ethically rigorous research with conclusions that are appropriately drawn based on the data presented.

Reviewer #1: Yes

Reviewer #2: Yes

Reviewer #3: No

2. Has the statistical analysis been performed appropriately and rigorously?

Reviewer #1: Yes

Reviewer #2: Yes

Reviewer #3: No

3. Have the authors made all data underlying the findings in their manuscript fully available (please refer to the Data Availability Statement at the start of the manuscript PDF file)?

Reviewer #1: Yes

Reviewer #2: Yes

Reviewer #3: Yes

4. Is the manuscript presented in an intelligible fashion and written in standard English?

Reviewer #1: Yes

Reviewer #2: Yes

Reviewer #3: Yes

5. Review Comments to the Author

Reviewer #1: Dear Author,

The methodology is vague with insufficient description of Introduction,. The result needs to be supported by more figures and tables. What is the relevence of author summary? the discussion is infufficinet to conclude your findings

Reviewer #2: The paper effectively introduces the purpose and methodology of a datathon focused on identifying biases in medical machine learning models. While some parts could benefit from improved fluency through the use of semicolons and more concise lines, the overall readability and flow, especially in the introduction, could be enhanced by reducing wordiness and redundancy. The authors have successfully demonstrated how a collaborative and interdisciplinary approach can uncover potential sources of bias, fostering a deeper understanding among participants. The inclusion of diverse participants and the use of various datasets and analytical tools further strengthen the study's findings. It would be beneficial to apply this methodology to larger and more diverse groups to address the critical issue of bias in medical AI.

Reviewer #3: Title: Raising awareness of potential biases in medical machine learning: Experience from a Datathon

Thank you for the opportunity to review this interesting piece of work. Find my comments hereunder.

Overview

The paper addresses the critical issue of potential biases in medical ML by enhancing users’ awareness of such issues through participation in a datathon. While AI is helpful in healthcare, there are still issues that need to be addressed, such as bias, which is often hidden in the blackbox nature of AI models, hence, the rise of XAI in healthcare. Thus, identifying and letting people know of biases is an important endeavour. However, the paper needs substantial improvements. For example, there is no literature review, which is critical in any research paper.

Introduction

Bias is a huge issue, and the introduction could be expanded to provide more context and a strong argument for bias avoidance and awareness.

Literature review

Provide a clearly delineated literature review. While it is possible to not have a dedicated section on literature review or related works following the IMRaD structure, the introduction should be detailed and long enough to provide a review of the literature, which isn’t the case in this manuscript.

Materials and methods

This section needs to be revised. E.g. The statement “Although logistic regression models based on demographics, labs, vitals, and APACHE score predictions led to models with good performance and calibration [7], potential biases in the data have not yet been thoroughly studied” is more of a research gap than methodology, especially the part “..potential biases in the data have not yet been thoroughly studied”. Describe how the submissions were analysed.

Results

The statement “Sixteen participants were assigned into five teams of equal size …” needs to be looked into, as 16 is indivisible by 5, hence, the teams can’t be of equal size. Again, if there were 5 equal teams out of 16 participants, team sizes cannot be 5, hence the statement “…final team sizes ranged from 1 to 5 members” needs to be corrected.

This part is more of methodology than results, as it describe ‘how’ the datathon was conducted:

“Of the five teams, two used the synthetic UPMC dataset, two used the women in science dataset, and one team used both. No teams used the eICU-CRD dataset Python was the preferred analysis tool, selected by four of the five teams. The remaining team used R. At least one team used a GUI-based tool for exploratory data analysis, and one team acknowledged using ChatGPT to assist with coding”

Discussion

Table 2 might have been misplaced, as it is cross-referenced in results but placed under the discussion section.

6. PLOS authors have the option to publish the peer review history of their article (what does this mean?). If published, this will include your full peer review and any attached files.

**Do you want your identity to be public for this peer review?** For information about this choice, including consent withdrawal, please see our Privacy Policy.

Reviewer #1: No

Reviewer #2: No

Reviewer #3: No

---

## [Decision Letter · Decision Letter 1]

14 May 2025

PDIG-D-24-00484R1Raising awareness of potential biases in medical machine learning: Experience from a DatathonPLOS Digital Health Dear Dr. Hochheiser, Thank you for submitting your manuscript to PLOS Digital Health. After careful consideration, we feel that it has merit but does not fully meet PLOS Digital Health's publication criteria as it currently stands. Therefore, we invite you to submit a revised version of the manuscript that addresses the points raised during the review process. Please submit your revised manuscript within 30 days Jun 13 2025 11:59PM. If you will need more time than this to complete your revisions, please reply to this message or contact the journal office at digitalhealth@plos.org. Please include the following items when submitting your revised manuscript: * A rebuttal letter that responds to each point raised by the editor and reviewer(s). You should upload this letter as a separate file labeled 'Response to Reviewers'. This file does not need to include responses to any formatting updates and technical items listed in the 'Journal Requirements' section below.* A marked-up copy of your manuscript that highlights changes made to the original version. You should upload this as a separate file labeled 'Revised Manuscript with Track Changes'.* An unmarked version of your revised paper without tracked changes. You should upload this as a separate file labeled 'Manuscript'. If you would like to make changes to your financial disclosure, competing interests statement, or data availability statement, please make these updates within the submission form at the time of resubmission. Guidelines for resubmitting your figure files are available below the reviewer comments at the end of this letter. We look forward to receiving your revised manuscript. Kind regards, Wisit Cheungpasitporn, MDAcademic EditorPLOS Digital Health Wisit CheungpasitpornAcademic EditorPLOS Digital Health Leo Anthony CeliEditor-in-ChiefPLOS Digital Healthorcid.org/0000-0001-6712-6626 **Additional Editor Comments (if provided):****Reviewers' Comments:**Reviewer's Responses to Questions

**Comments to the Author**

1. If the authors have adequately addressed your comments raised in a previous round of review and you feel that this manuscript is now acceptable for publication, you may indicate that here to bypass the “Comments to the Author” section, enter your conflict of interest statement in the “Confidential to Editor” section, and submit your "Accept" recommendation.

Reviewer #1: All comments have been addressed

Reviewer #3: (No Response)

Reviewer #4: All comments have been addressed

2. Does this manuscript meet PLOS Digital Health’s publication criteria? Is the manuscript technically sound, and do the data support the conclusions? The manuscript must describe methodologically and ethically rigorous research with conclusions that are appropriately drawn based on the data presented.

Reviewer #1: Partly

Reviewer #3: Yes

Reviewer #4: Yes

3. Has the statistical analysis been performed appropriately and rigorously?

Reviewer #1: I don't know

Reviewer #3: N/A

Reviewer #4: Yes

4. Have the authors made all data underlying the findings in their manuscript fully available (please refer to the Data Availability Statement at the start of the manuscript PDF file)?

Reviewer #1: No

Reviewer #3: Yes

Reviewer #4: Yes

5. Is the manuscript presented in an intelligible fashion and written in standard English?

Reviewer #1: Yes

Reviewer #3: No

Reviewer #4: Yes

6. Review Comments to the Author

Reviewer #1: Dear Author,

I comply with my previous decision

Reviewer #3: Please see the attached file for comments

Reviewer #4: In Results section, two instances refer to “table ??”—a placeholder likely meant to be replaced with correct table number. This may confuse readers and reduce manuscript’s polish. Authors should ensure all tables are consistently and accurately numbered (e.g., “Table 1”) and cited in text. Clear and precise cross-referencing enhances readability and professionalism of manuscript.

In Introduction (line 23), a typographical error appears in phrase “race as in a predicto,” which seems either truncated or mistakenly constructed. Intended meaning was likely to describe use of race as predictive variable in algorithms. Sentence should be revised for clarity to: “Many algorithms have explicitly used race as predictive variable.” Correction will improve clarity and restore grammatical accuracy.

Dataset is referred to inconsistently, including “women in science dataset” and “Women in Data Science Datathon 2020 dataset.” Standardizing dataset name throughout text is recommended. Using consistent term such as “Women in Data Science (WiDS) 2020 dataset” will prevent confusion and ensure precision, especially for readers unfamiliar with dataset.

Results note that only one team employed inferential statistics during modeling process. While observation is included, manuscript does not reflect on its implications. Including a brief comment in Discussion to acknowledge this limitation would strengthen manuscript’s critical analysis. Authors may note that limited use of statistical testing highlights need for stronger emphasis on methodological rigor in data science education formats like Datathons.

In Discussion (lines 182–183), sentence “Anecdotal observation of groups in action suggested that exchanges between clinicians and informaticians helped participants better appreciate...” reflects inconsistent verb tense. For consistency and readability, sentence should be revised fully in past tense—for example: “Anecdotal observations... suggested that exchanges enhanced participants’ understanding.” Maintaining uniform verb tense supports coherent narrative flow in discussion.

7. PLOS authors have the option to publish the peer review history of their article (what does this mean?). If published, this will include your full peer review and any attached files.

**Do you want your identity to be public for this peer review?** For information about this choice, including consent withdrawal, please see our Privacy Policy.

Reviewer #1: None

Reviewer #3: No

Reviewer #4: No

**Figure resubmission:**While revising your submission, please upload your figure files to the Preflight Analysis and Conversion Engine (PACE) digital diagnostic tool, https://pacev2.apexcovantage.com/. PACE helps ensure that figures meet PLOS requirements. To use PACE, you must first register as a user. Registration is free. Then, login and navigate to the UPLOAD tab, where you will find detailed instructions on how to use the tool. If you encounter any issues or have any questions when using PACE, please email PLOS at figures@plos.org. Please note that Supporting Information files do not need this step. If there are other versions of figure files still present in your submission file inventory at resubmission, please replace them with the PACE-processed versions.**Reproducibility:**To enhance the reproducibility of your results, we recommend that authors of applicable studies deposit laboratory protocols in protocols.io, where a protocol can be assigned its own identifier (DOI) such that it can be cited independently in the future. Additionally, PLOS ONE offers an option to publish peer-reviewed clinical study protocols. Read more information on sharing protocols at https://plos.org/protocols?utm_medium=editorial-email&utm_source=authorletters&utm_campaign=protocols

---

## [Decision Letter · Decision Letter 2]

23 Jun 2025

Raising awareness of potential biases in medical machine learning: Experience from a Datathon

PDIG-D-24-00484R2

Dear Hochheiser,

We are pleased to inform you that your manuscript 'Raising awareness of potential biases in medical machine learning: Experience from a Datathon' has been provisionally accepted for publication in PLOS Digital Health.

Best regards,

Wisit Cheungpasitporn, MD

Academic Editor

PLOS Digital Health

**Additional Editor Comments (if provided):**

I have reviewed the revised manuscript and the authors’ responses to the reviewers' comments. All concerns have been adequately addressed. I have no further comments and recommend acceptance.

**Reviewer Comments (if any, and for reference):**

Reviewer's Responses to Questions

**Comments to the Author**

1. If the authors have adequately addressed your comments raised in a previous round of review and you feel that this manuscript is now acceptable for publication, you may indicate that here to bypass the “Comments to the Author” section, enter your conflict of interest statement in the “Confidential to Editor” section, and submit your "Accept" recommendation.

Reviewer #3: All comments have been addressed

2. Does this manuscript meet PLOS Digital Health’s publication criteria? Is the manuscript technically sound, and do the data support the conclusions? The manuscript must describe methodologically and ethically rigorous research with conclusions that are appropriately drawn based on the data presented.

Reviewer #3: (No Response)

3. Has the statistical analysis been performed appropriately and rigorously?

Reviewer #3: (No Response)

4. Have the authors made all data underlying the findings in their manuscript fully available (please refer to the Data Availability Statement at the start of the manuscript PDF file)?

Reviewer #3: (No Response)

5. Is the manuscript presented in an intelligible fashion and written in standard English?

Reviewer #3: (No Response)

6. Review Comments to the Author

Reviewer #3: My comments were addressed

7. PLOS authors have the option to publish the peer review history of their article (what does this mean?). If published, this will include your full peer review and any attached files.

**Do you want your identity to be public for this peer review?** For information about this choice, including consent withdrawal, please see our Privacy Policy.

Reviewer #3: No
